# Spatiotemporal inflection points in human running: Effects of training level and athletic modality

**Yuta Goto[1], Tetsuya Ogawa[2], Gaku Kakehata[1], Naoya Sazuka[3], Atsushi Okubo[4], Yoshihiro Wakita[4], Shigeo Iso[5], Kazuyuki Kanosue[5]***

**1** Graduate School of Sport Sciences, Waseda University, Saitama, Japan, **2** Department of Clothing, Faculty of Human Sciences and Design, Women's University Tokyo, Japan, **3** Tokyo Laboratory 25, R&D Center, Sony Group Corporation, Tokyo, Japan, **4** Tokyo Laboratory 07, R&D Center, Sony Group Corporation, Tokyo, Japan, **5** Faculty of Sport Sciences, Waseda University, Saitama, Japan

* kanosue@waseda.jp

**Data Availability Statement:** All relevant data are within the manuscript and its Supporting Information files.

## Abstract

The effect of the different training regimes and histories on the spatiotemporal characteristics of human running was evaluated in four groups of subjects who had different histories of engagement in running-specific training; sprinters, distance runners, active athletes, and sedentary individuals. Subjects ran at a variety of velocities, ranging from slowest to fastest, over 30 trials in a random order. Group averages of maximal running velocities, ranked from fastest to slowest, were: sprinters, distance runners, active athletes, and sedentary individuals. The velocity-cadence-step length (V-C-S) relationship, made by plotting step length against cadence at each velocity tested, was analyzed with the segmented regression method, utilizing two regression lines. In all subject groups, there was a critical velocity, defined as the inflection point, in the relationship. In the velocity ranges below and above the inflection point (slower and faster velocity ranges), velocity was modulated primarily by altering step length and by altering cadence, respectively. This pattern was commonly observed in all four groups, not only in sprinters and distance runners, as has already been reported, but also in active athletes and sedentary individuals. This pattern may reflect an energy saving strategy. When the data from all groups were combined, there were significant correlations between maximal running velocity and both running velocity and step length at the inflection point. In spite of the wide variety of athletic experience of the subjects, as well as their maximum running velocities, the inflection point appeared at a similar cadence (3.0 ± 0.2 steps/s) and at a similar relative velocity (65–70%Vmax). These results imply that the influence of running-specific training on the inflection point is minimal.

## Introduction

Human running has been studied extensively from the viewpoint of how its temporal (cadence) and spatial (step length) components contribute to velocity [1–10]. Velocity equals the product of cadence and step length, and the relative contribution of each component to

**Funding:** This work was supported by Japan Society for the Promotion of Science (JSPS), KAKENHI Grant Number 19K22822 (K.K) and by Grant-in-Aid for JSPS Fellows Number 20J11122 (Y.G) from Ministry of Education, Culture, Sports, Science and Technology of Japan. Sony Group Corporation provided support in the form of salaries for authors [NS, AO, and YW], but did not have any additional role in the study design, data collection and analysis, decision to publish, or preparation of the manuscript. The specific roles of these authors are articulated in the 'author contributions' section. Sony Group Corporation has a patent (US20180039751A1) on apparatuses for helping runners modify the V-C-S property. This patent does not interfere with the usage of any data or knowledge presented in the paper.

**Competing interests:** Sony Group Corporation provided support in the form of salaries for authors [NS, AO, and YW].Sony Group Corporation does not alter the adherence to PLOS ONE policies on sharing data and materials presented in this paper.

changing velocity differs across the velocity range. A previous study reported that, at slower velocities, speed is modulated primarily by adjusting step length, whereas, at faster velocities, speed is modulated more by changes in cadence [6]. At velocities close to maximum, step length shows only a small increase or even a decrease as running velocity approaches the maximum [8]. These characteristics are considered to indicate the spontaneous recruitment of an adequate motor pattern which minimizes energy expenditure at a given running velocity [5, 11–13]. Mechanical approaches, such as Fenn's approach, have been used as useful tools to elucidate these energy cost determinants with many practical applications [14]. Yanai and Hay [12], utilizing a two-dimensional simulation, evaluated the relative contribution of cadence and step length in the optimization of power production utilizing both anatomical (range of motion in the hip joint) and spatiotemporal (duration of the stance phase) determinants. Indeed, if the cadence is voluntarily modified from that occurring under the natural movement pattern at a given running velocity, metabolic rate is lowest when the cadence is in the range of ±10% of the preferred cadence [15–18]. In addition, in the slower velocity range, Cavagna et al. [19] reported that preferred cadences take place in the proximity of 3 Hz.

However, the extent to which the above characteristics occur in different populations and in persons with different physical backgrounds remains unclear. Most of the above-mentioned studies focused on well-trained individuals, especially those trained for running [7–9, 12, 20].

Therefore, the purpose of the present study was to investigate how a change in running velocity altered the spatiotemporal adjustment between step length and cadence in subjects with different histories of engagement in running training. Namely, we studied:1. sprinters, 2. distance runners, 3. active athletes who had received no running-specific training, and 4. sedentary, untrained subjects. The relationships among running velocity, cadence, and step length over a wide range of running velocities were compared across these subjects. Among the four groups, the distance runners would be expected to run as efficiently (either mechanically or metabolically) as possible. As noted above, in the slower velocity ranges, altering stride length is a more energy saving strategy for changing velocity than is altering cadence [12]. Therefore, we hypothesized: 1. the running step length/cadence patterns of individuals would be influenced by their running training experience and overall physical activity levels and 2. distance runners would exhibit the greatest tendency to change velocity by altering step length in the slower velocity range.

## Methods

### Subjects

A total of eighty volunteers (69 males and 11 females) with different backgrounds, in terms of their running experience, participated in the study. They were assigned into one of four groups depending on their current/previous running training. We utilized four groups of subjects with different histories of running training. The first and second groups consisted of twenty sprinters (all men) and twenty distance runners (all men), respectively. The participants in the third group were twenty active athletes (16 males and 4 females). Although running is involved in many of the sports, all subjects informed us that they had received no special training for improving their running speed. For reference, the sports that the participants in the third group engaged in were: soccer, basketball, softball, weightlifting, boxing, lacrosse, volleyball, American football, badminton, handball, rowing, judo, and golf. They had all participated in their sport for at least 5 years. The fourth group consisted of sedentary individuals without a history of any regular participation in sports activities (13 males and 7 females). Table 1 lists the characteristics of participants in each group. All participants were informed of the purposes and procedures, and signed an informed consent form. This study was approved by the

**Table 1. Physical characteristics and sport activity history of each subject group.**

|  | N | age, years | height, cm | sports activity history, years |
|---|---|---|---|---|
| Sprinters | 20 | 22 ± 2 | 176.2 ± 6.1[b, c, d] | 9.7 ± 3.0 |
| Distance runners | 20 | 20 ± 1 | 171.0 ± 4.5 | 7.4 ± 2.0 |
| Active athletes | 20 | 23 ± 2 | 170.1 ± 5.8 | 10.2 ± 4.4 |
| Sedentary individuals | 17 | 22 ± 2 | 166.0 ± 6.2 |  |

Values are means ± SD. N, number of subjects. b, c, d: values are significantly different from distance runners, active athletes, and sedentary individuals, respectively ($p < 0.05$). The sport activity history of the active athletes indicates the number of years of participation in that sport for each subject.

Human Research Ethics Committee in Faculty of Sport Sciences, Waseda University. The experiments were conducted in accordance with the Declaration of Helsinki.

## Experimental setup and tasks

Experiments were conducted on a 30 m all-weather straight track (only 20 m for the sedentary group in consideration of their physical strength and lack of stamina) on which color markers were placed every 0.5 m for video analysis. A sagittal view of each participant was recorded by panning with a video camera (HDR-CX630V, SONY) placed approximately 10 m lateral to the center of the running path. An additional 10–30 meters was provided before and after the filming zone (of 30m or 20m) so that the subjects could accelerate and decelerate and thus maintain running velocity as constant as possible throughout the recording area. This acceleration distance differed between trials and was selected by the subject. The video sampling frequency was 60 Hz.

Participants were asked to run along the path 30 times at a variety of velocities, which varied from slow to the fastest possible. The order of running with different velocities was randomized on a subject-by-subject basis. The subjects were directed to run at a particular percentage of their maximal effort [21]. This instruction included requesting a subjective effort from 10% to 100% of maximum, as well as "run faster or slower than the previous trial". The actual running speed did not necessarily match the exact percentage of their maximal speed. However, this method did produce the necessary array of running speeds and the subjects might run more than once at an intensity. When running at the minimum velocity, subjects followed our instruction to run as slowly as they could while still maintaining a running gait (as opposed to walking, jumping, hopping, or bounding). The interval between trials ranged from 30 seconds to 5 minutes, depending on the speed of the previous trial. A 5-minute rest was taken after 15 trials. The participants used their own running shoes. Spiked shoes were not allowed.

## Data analysis

Offline data analysis was performed by using video administration software (PlayMemories, SONY, Japan). On the basis of the video analysis, the running velocity, cadence, and step length were calculated on a trial-by-trial basis for each subject. Mean running velocity (m/sec) was calculated by dividing the length of the path (m) by the time taken (sec) to run over the path. The instant at which the subject passed the start and the end point were identified from the position of the chest relative to the color markers. Mean cadence (steps/sec) was calculated by dividing the number of steps by the time taken to cover that distance. The number of steps was counted from the first ground contact with the path to the last ground contact before passing the end point. The duration utilized was defined as the time between the instant of first foot-contact after the start position and that of the last foot-contact before the end. Mean step length (m) was calculated by dividing the mean running velocity (m/sec) by the mean cadence (steps/sec). Step length was also

expressed as the ratio of the step length (m) to the height (m) of each subject in order to examine the influence of the physical characteristics of the subjects. For the running velocity, the fastest among the 30 trials by each subject was designated as their maximal running velocity.

In the present study, the principal analyses for the spatiotemporal running characteristics of each subject were performed with MATLAB version R2018a (The MathWorks, Inc., USA). For each subject, the data were plotted as shown in Fig 1 in order to examine the relationship between cadence and step length (horizontal axis: cadence, vertical axis: step length). This correspondence involved the Velocity (m/s, dotted line), Cadence (steps/s, horizontal), and Step length (m, vertical), and is defined as the V-C-S relationship. To quantitatively analyze the critical point at which the relative contribution of spatiotemporal adjustment changed (cadence vs. step length), we utilized the segmented regression method which has previously been used to detect lactate threshold [22] and ventilation threshold [23] during aerobic exercise. This is a statistical method for determining the point at which a line suddenly changes slope at some unknown point. We used a segmented regression procedure [23, 24] in which the N data points were divided into two segments (the lower x data and the upper N-x data, x = 3, 4, . . ., or N-2). Each segment was fitted with a regression line using the Deming regression [25, 26]. This regression method was adopted to exclude the effects of measurement errors in cadence and step length. That is, one regression line was obtained with x data points from the ascending order starting with the minimum velocity, and the other one with N-x data points from the descending order starting with the maximum velocity. The critical point ("inflection point"), then, was the intersection of the two regression lines with an x value that minimized orthogonal distance between measurement data and regression line for two data sets (segments) (Fig 1, cross; X). We assumed that the regression lines below and above the inflection point would adequately represent the spatiotemporal characteristics of running for each subject and group.

Subjects with inflection points, thus obtained, that differed largely from the measured points, were excluded from the analysis (#18, #19, and #20, as seen in S4 Fig).

Therefore, the final analysis involved 20 sprinters, 20 distance runners, 20 active athletes, and 17 sedentary individuals. For these subjects, running velocity, cadence, and step length at

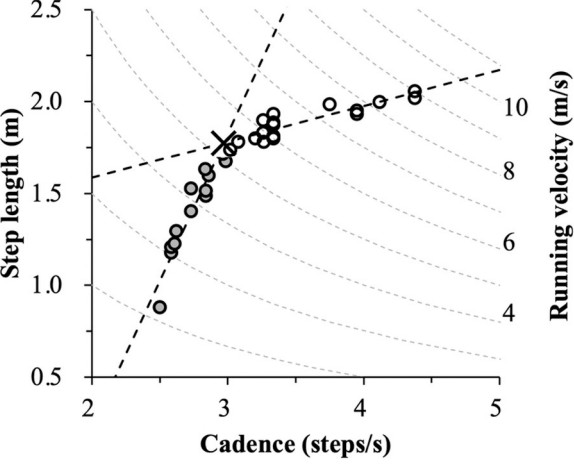

**Fig 1. The relationship between cadence (steps/s, horizontal) and step length (m, vertical) relative to running velocity (pale broken line and the second vertical axis) in a single sprinter.** The inflection point (cross) was computed from two regression lines from different data sets by combining the segmented regression method of Deming regression. The filled and open circle markers represent the data sets below and above the inflection point at which the relationship between cadence and step length changed abruptly. Inflection point was obtained as the intersection point of the two regression lines.

the inflection point were calculated. Normalized values were determined for each parameter at the maximal running velocity.

## Statistical analysis

Statistical analysis was performed using SPSS Statistics 23 software (IBM, USA). Maximal running velocity, height of subjects, and all variables related to inflection point in each group were tested for a normal distribution using the Shapiro-Wilk test. Maximal running velocity, height, and normalized cadence at the inflection point were found to have non-normal distributions. Thus, group mean data for maximal running velocity, height of subjects, and all variables related to inflection point were analyzed among the four subject groups by using a non-parametric Kruskal-Wallis test. Next, post-hoc pairwise comparisons using the Dunn-Bonferroni approach were made to identify additional differences between the groups. In order to further investigate the possible mechanisms responsible for the inflection point, correlational analyses were performed.

All variables across all subjects related to the inflection point and maximal running velocity were tested for a normal distribution using the Shapiro-Wilk test. Maximum running velocity, and step length at maximal running velocity exhibited normal distributions. Likewise, running velocity (both unnormalized and normalized), step length (both unnormalized and normalized), and unnormalized cadence at the inflection point exhibited normal distributions. However, cadence at maximal running velocity and normalized cadence at the inflection point exhibited non-normal distributions. Pearson's and Spearman's correlations were performed to analyze the relationship between maximal running velocity and other parameters at the inflection point. Significance was set at $p < 0.05$. The data are presented as mean and standard deviation (mean ± SD).

## Results

Fig 1 shows a typical example of the relationship between running velocity, cadence, and step length for a single sprinter. Both cadence and step length show specific changes in relation to changing running velocity. The inflection point (cadence: 2.97 steps/s, step length: 1.78 m) was computed from two regression lines.

Fig 2A shows an inter-group comparison of the mean values of Vmax. A Kruskal-Wallis test revealed significant differences between the groups in terms of maximum running velocity ($\chi^2$ (3) = 52.463, p < 0.001). The post-hoc comparisons revealed that the maximal velocity of the sprinters was faster compared to all the other subject groups (distance runner: p = 0.009, active athlete: p < 0.001, sedentary: p < 0.001). The distance runner group exhibited significantly faster maximal running velocity in comparison with the sedentary individual group. Fig 2B–2D illustrates the correlation between maximal running velocity and cadence, absolute step length and step length normalized to height at the maximal running velocity. There were significant positive correlations between Vmax and cadence as well as step length both in the unnormalized and normalized forms (cadence: r = 0.514, p < 0.001; step length (unnormalized): r = 0.843, p < 0.001; step length (normalized): r = 0.803, p < 0.001).

Fig 3A shows mean values of cadence and step length at maximal running velocity (Vmax), the inflection point, and minimal running velocity (Vmin) for each subject group. As shown in Fig 3A, maximal running velocity was different across the groups and was the fastest in the sprinters (I, around 10 m/s) and slowest in the sedentary individuals (IV, mostly less than 8 m/ s). All groups tended to increase step length predominately at the velocities between Vmin (velocity: 2.17 ± 0.45 m/s, cadence: 2.62 ± 0.14 steps/s, step length: 0.82 ± 0.17 m) and the inflection point, and then to increase cadence until they reached Vmax. Fig 3B depicts mean values of cadence and step length normalized to the values obtained under maximal running

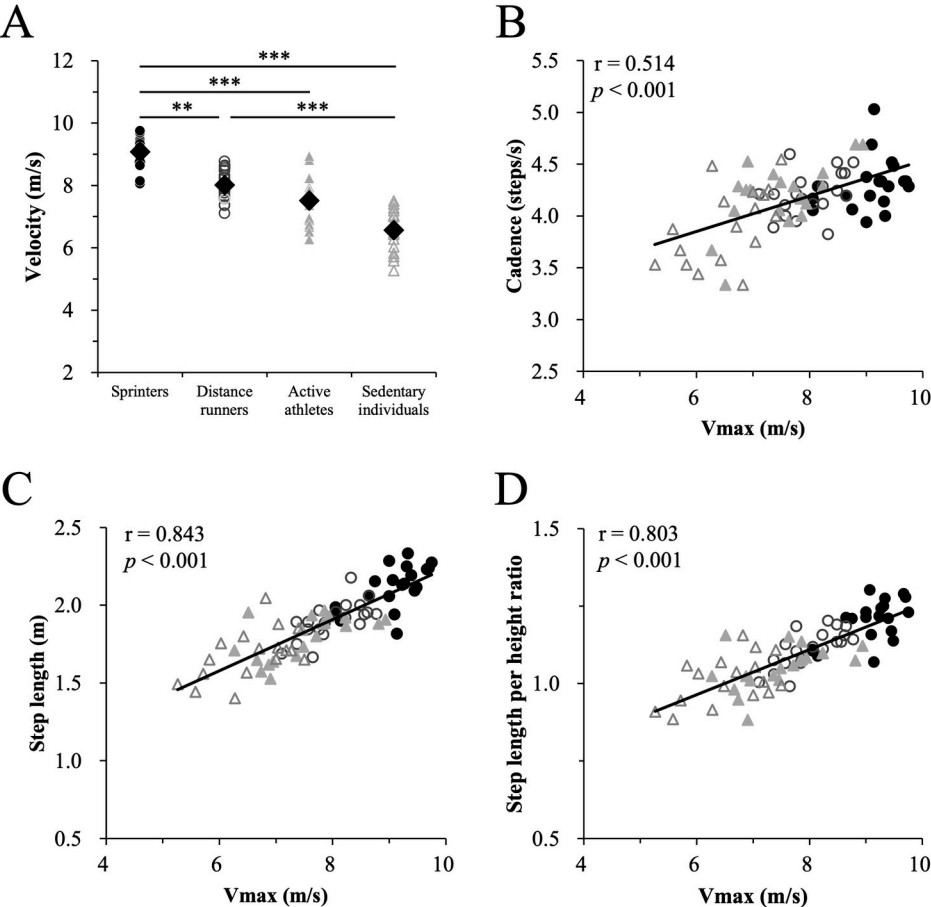

**Fig 2.** Inter-group comparison of mean values (diamond) of the maximum running velocity (Vmax) (A), and correlation between the maximal running velocity and the cadence (B), step length (C), and step length normalized by height (D) at maximal running velocity. In Fig 2A, open circles indicate each individual subject. Significant difference; ***p < 0.001, **p < 0.01. In Fig 2B–2D, filled circles, open circles, filled triangles, and open triangles represent the sprinters, distance runners, active athletes, and sedentary individuals, respectively. There are significant positive correlations between Vmax and the cadence (B) and between Vmax and step length, both absolute velocity and velocity normalized to maximal running velocity (r = 0.514, p < 0.001; r = 0.843, p < 0.001; r = 0.803, p < 0.001, respectively).

velocity. The characteristics of the increase in velocity were similar to those from Fig 3A. Due to differences in the absolute value (Fig 3A) of maximal running velocity, the normalized cadence varied considerably across the subject groups, while variability in step length below the inflection point was less evident.

Table 2 shows inter-group comparison of the mean values of all variables related to the inflection point. A Kruskal-Wallis test revealed significant difference of running velocity, step length, normalized cadence ($\chi^2$ (3) = 31.215, p < 0.001; $\chi^2$ (3) = 42.68, p < 0.001; $\chi^2$ (3) = 23.623, p < 0.001, respectively). The post-hoc comparisons revealed significant differences between the subject groups. In the group of sprinters, the running velocity was significantly faster as compared to the active athlete, and sedentary subject groups (active athlete: p < 0.01, sedentary: p < 0.001). For the same parameter, the group of distance runners showed significantly faster in comparison to the sedentary group (p < 0.01). The step length was significantly longer in the sprinter group in comparison to all the other subject groups (distance runner: p < 0.01, active athletes: p < 0.001, sedentary: p < 0.001). For the same parameter, the group of distance runners was significantly longer than the sedentary group (p < 0.05). In the group

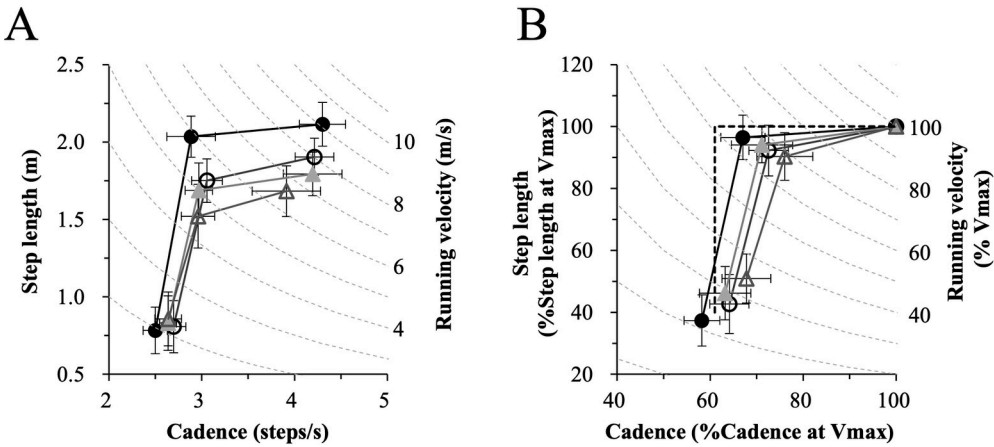

**Fig 3.** Mean values of cadence and step length at the maximal running velocity (Vmax), inflection point (IP), and minimal running velocity (Vmin) (A), and those with cadence and step length normalized to those under Vmax (B) for each subject group. The error bars depict the standard deviation. The filled circles, open circles, filled triangles, and open triangles represent sprinters, distance runners, active athletes and sedentary individuals, respectively. Pale broken lines represent running velocity (A) and running velocity normalized by maximal running velocity (B). The thick broken line in B illustrates the limiting situation, in which velocity change is only done with a step length change in the velocity range below the inflection point, and only with a cadence change above the inflection point.

of sprinters, the normalized cadence was lower as compared to distance runner and sedentary subject groups (distance runner: $p < 0.01$, sedentary: $p < 0.001$).

Fig 4A–4C depicts correlations between maximal running velocity and running velocity, cadence, and step length at the inflection point. There were significant positive correlations between Vmax and both velocity and step length at the inflection point (velocity: Fig 4A, $r = 0.738$, $p < 0.001$; step length: Fig 4C, $r = 0.827$, $p < 0.001$). Cadence at the inflection point had no correlation with Vmax, and was approximately constant at $3.0 \pm 0.2$ steps/s regardless of the subject group (Fig 4B). Fig 4D–4F illustrates correlation for the same parameters shown in Fig 4A–4C, but with values normalized to Vmax. Velocity and cadence show negative correlations (velocity: $r = -0.300$, $p < 0.01$; cadence: $r = -0.621$, $p < 0.001$), while step length has a positive correlation with Vmax ($r = 0.290$, $p < 0.05$).

## Discussion

We investigated the relative contribution of cadence and step length changes as running velocity was modulated in four groups of subjects with different histories of engagement in

**Table 2. Kinematic variables at the inflection point.**

|  | Sprinters (N = 20) | Distance runners (N = 20) | Active athletes (N = 20) | Sedentary individuals (N = 17) |
|---|---|---|---|---|
| velocity, m/s | 5.86 ± 0.59[c, d] | 5.36 ± 0.60 [d] | 5.00 ± 0.50 | 4.50 ± 0.65 |
| step length, m | 2.03 ± 0.13 [b, c, d] | 1.75 ± 0.14 [d] | 1.69 ± 0.18 | 1.52 ± 0.21 |
| cadence, steps/s | 2.88 ± 0.26 | 3.06 ± 0.17 | 2.97 ± 0.15 | 2.96 ± 0.18 |
| normalized velocity, % | 64.7 ± 7.1 | 67.0 ± 7.5 | 66.7 ± 4.8 | 68.6 ± 7.4 |
| normalized step length, % | 96.5 ± 7.2 | 92.2 ± 8.2 | 94.1 ± 5.8 | 90.3 ± 7.7 |
| normalized cadence, % | 67.0 ± 4.7 [b, d] | 72.6 ± 4.2 | 71.2 ± 6.6 | 76.0 ± 6.0 |

Values are means ± SD. N, number of subjects. b, c, d: values are significantly larger, from distance runners, active athletes, and sedentary individuals, respectively. Normalized velocity, step length, and cadence were obtained by normalizing with corresponding values at the maximal running velocity, respectively.

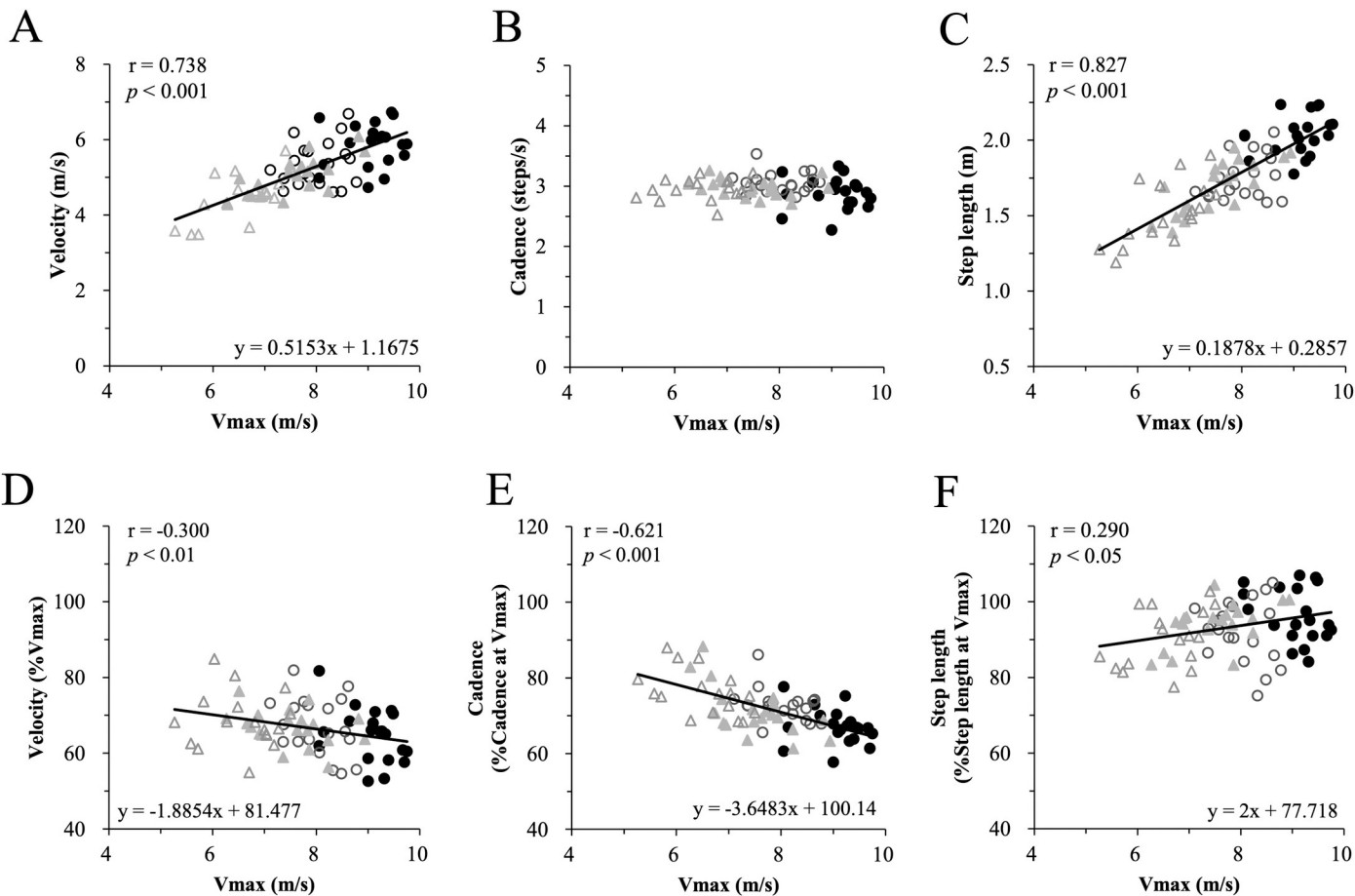

**Fig 4.** Correlation between maximal running velocity (Vmax) and: running velocity (A), cadence (B), and step length (C), as well as the same three parameters normalized to the Vmax (D–F) at the inflection point. Filled and open circles, and filled and open triangles represent the sprinters, distance runners, active athletes, and sedentary individuals, respectively. The correlations are all significant except for cadence (B).

running-specific training, utilizing the segmented regression method with two regression lines (Fig 1). In spite of a large variation in maximal running velocity, the general characteristics of the V-C-S relationship were similar across the subject groups (Fig 3) as well as across the data of individuals (S1–S4 Figs).

## Basic characteristics of the V-C-S relationship

As expected, compared to the sprinters, maximal running velocities were progressively slower in the distance runners, active athletes and sedentary groups. There were significant differences between the sprinters and the other three groups, as well as between the distance runners and the sedentary individuals (Fig 2A). Both cadence and step length at Vmax were well correlated with Vmax (Fig 2B and 2C, respectively). Among the subject groups, the sprinters were the tallest and the sedentary group was the shortest. The strong correlation of step length with Vmax was well-preserved, however, even when step length was normalized to the subjects' heights (Fig 2D). Thus, faster maximum running velocities were generally accomplished with both a higher cadence and longer steps. The minimum running velocity was common to all subject groups at 2.17 ± 0.45 m/s with a cadence of 2.62 ± 0.14 steps/s and a step length of

0.82 ± 0.17 m (Fig 3A). It appears that a slower cadence would have required "hopping" rather than running, and for shorter step lengths it became similar to "jogging in place".

In all four subject groups, an abrupt change in the V-C-S relationship took place at the inflection point (Fig 3 and Table 2). Velocity changes below the inflection point occurred mainly by modulating step length and velocity changes above the inflection point occurred mainly via cadence modulation. These characteristics were demonstrated in preceding studies conducted on sprinters and distance runners [7, 9], and are particularly prominent in sprinters.

Running velocity at the inflection point has a significant positive correlation with Vmax (Fig 4A). Thus, the faster the Vmax, the faster the velocity at the inflection point. A faster velocity at the inflection point is mainly attained by longer step length (Fig 4C). However, this correlation was weak when it is normalized with the step length at the Vmax (Fig 4F).

Overall, regardless of the training history, all groups had a similar relative step length quite close to the maximum step length (about 90%). Interestingly, the cadence at the inflection point has no correlation with Vmax and remained constant at about 3 steps/sec (Fig 4B). The history of the training influenced normalized cadence at the inflection point, that is, sprinters had a lower normalized cadence at the inflection point than the others, although in absolute terms cadence was the same. In the normalized plane (Fig 3B) inflection points of the different groups are lined along the isovelocity curve of 65–70%. Scatter plots of all subjects of all the groups showed only a weak correlation between the Vmax and the velocity at the inflection point normalized with Vmax (Fig 4D). In spite of the wide range of sports, and thus athletic modality of the subjects as well as their maximum running velocity, the inflection point appeared at a similar cadence (3.0 ± 0.2 steps/s) as well as at similar relative velocity (65–70% Vmax), across all groups. These results imply that the influence of running-specific training on the inflection point is minimal.

## Functional meaning of the V-C-S relationship

Although the basic characteristics of the V-C-S relationship are common across different subject groups, the quantitative difference could be related to quality/quantity difference in running-specific training among groups.

In the present study, four groups of subjects, sprinters, distance runners, active athletes utilizing varying degrees of running but no running training, and sedentary individuals, were studied. Of course, the above order would also be expected for the maximal velocity from fastest to the slowest (Fig 2A). Sprinting and distance training involves running on a daily basis, and running (generally without specific running instruction) forms one aspect of training for many of the active athletes as well. It seems reasonable that some portion of the observed maximal velocities reflect differences in training.

Interestingly, step length at the inflection point also follows the same order as the maximal velocity (Figs 3A and 4C and 4F). In the velocity range below the inflection point, velocity change is mainly done with a change in step length; for energy-saving this is a more efficient strategy than is changing the cadence [12]. It would be beneficial for distance runners to run within this range as much as possible when their velocity is below the inflection point. Indeed, it was shown that at 4.4 m/s velocity, in the range below the inflection point, the stride length was associated with better running economy in distance runners [27]. Therefore, we had hypothesized that the ability to run below the inflection point would be particularly developed in distance runners. However, sprinters and not distance runners increased velocity by elongating both absolute step length (Fig 4C) and relative step length (Fig 4F), all the way to the upper running speed limit. Thus, our working hypothesis was rejected. Sprinters rarely train

in the velocity range below the inflection point. Obviously, maximal velocity is crucial for sprinters. A faster velocity cannot be accomplished only with power, especially at the highest levels. Sprinters need to develop both power and economy to the upper limit, and inevitably and unintentionally develop mechanically efficient movements.

### Future studies

Why and by what means are there differences in the various parameters of the V-C-S relationship? In particular, the neural as well as physiomechanical mechanisms of differences in the V-C-S relationship should prove very interesting. In the future, motion analysis together with measurements of muscle activity and ground reaction forces could help to answer our overall question. Although numerical simulation of running and walking has many limitations [11, 12, 28], the differences in the V-C-S relationship could be analyzed with numerical models in terms of various energy costs. Furthermore, it is very interesting that even in the sedentary subjects, the basic pattern of V-C-S relationship, which is considered to reflect efficiency [12, 13], was seen. Is the V-C-S pattern innate or does it develop along the development? This, and also fatigue [29], aging [30, 31], and sex differences [32], if any, are topics that merit future analysis.

### Conclusions

In the present study we analyzed the V-C-S relationship of running with the segmented regression method and made a quantitative comparison of the "spatiotemporal running characteristics" in subjects with different histories of running-specific training. The common characteristic of the V-C-S relationship is, in the slower and faster velocity ranges, that velocity is mainly modulated by altering step length and cadence, respectively. This was observed not only in the sprinters and distance runners, as shown in previous studies, but in active (general sport) athletes and sedentary subjects as well. In spite of the wide range of athletic modalities of the subjects, and their maximum running velocity, the inflection point appeared at a similar cadence (3.0 ± 0.2 steps/s) and at similar a relative velocity (65–70%Vmax), across all groups. These results imply that the influence of running-specific training on the inflection point is minimal.

### Supporting information

**S1 Fig. The relationship between cadence and step length for all the sprinters.** The two dashed lines depict the regression lines computed from different data below and above the inflection point, respectively.
(PDF)

**S2 Fig. The relationship between cadence and step length for all the distance runners.** The two dashed lines show the regression lines computed from different data below and above the inflection point, respectively.
(PDF)

**S3 Fig. The relationship between cadence and step length for the active athletes.** The two dashed lines show the regression lines computed from different data below and above the inflection point, respectively. The title of each figure corresponds to each subject's sports experience. Characters in parentheses signify male or female subjects.
(PDF)

**S4 Fig. The relationship between cadence and step length for the sedentary individuals.** The two dashed lines show the regression lines computed from different data below and above the inflection point, respectively. In the sedentary group, three subjects were excluded from

data analysis: two subjects (No. 18 and No. 19) had estimated inflection point fell outside the range of the original data, and one subject (No. 20) showed two regression lines with almost the same slope giving the inflection point completely outside the range of measured data. Characters in parentheses signify male or female subjects.
(PDF)

## Acknowledgments

The authors thank Dr. Larry Crawshaw for English editing of the manuscript.

## Author Contributions

**Conceptualization:** Yuta Goto, Tetsuya Ogawa, Gaku Kakehata, Kazuyuki Kanosue.

**Formal analysis:** Yuta Goto, Naoya Sazuka, Yoshihiro Wakita.

**Funding acquisition:** Yuta Goto.

**Investigation:** Yuta Goto, Gaku Kakehata.

**Methodology:** Yuta Goto, Tetsuya Ogawa, Naoya Sazuka, Yoshihiro Wakita.

**Project administration:** Yuta Goto, Atsushi Okubo, Kazuyuki Kanosue.

**Software:** Naoya Sazuka.

**Supervision:** Kazuyuki Kanosue.

**Visualization:** Yuta Goto.

**Writing – original draft:** Yuta Goto, Naoya Sazuka, Yoshihiro Wakita.

**Writing – review & editing:** Tetsuya Ogawa, Gaku Kakehata, Atsushi Okubo, Shigeo Iso, Kazuyuki Kanosue.

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
