## [Decision Letter · Decision Letter 0]

12 Apr 2021

PONE-D-21-04779

Effect of running-specific training on the spatiotemporal coordination of running in humans.

PLOS ONE

Dear Dr. Kanosue,

Thank you for submitting your manuscript to PLOS ONE. After careful consideration, we feel that it has merit but does not fully meet PLOS ONE’s publication criteria as it currently stands. Therefore, we invite you to submit a revised version of the manuscript that addresses the points raised during the review process.

There is a clear research question and you have applied the methods well to answer this question. The predominance of speed-dependent mechanical orchestration is a very interesting research question. This study provides information and reasoning that, while not exhausting, adds a clearer picture of this mechanical running orchestration. In general, what you need improve is (i) writing at various points in the manuscript; (ii) a physiomechanical justification for these specific adjustments (orchestration); (iii) a more informative and appealing title ... suggestion: Spatiotemporal inflection points in human running: effects of training level and athletic modality. Although the reviewers made different decisions (minor, major and reject), the opinions are univocal in stating that the study question and the method used are satisfactory and that modifications especially in the form of writing and in the deepening of some interpretations are needed.

We look forward to receiving your revised manuscript.

Kind regards,

Leonardo A. Peyré-Tartaruga, Ph.D.

Academic Editor

PLOS ONE

Journal Requirements:

We note that one or more of the authors are employed by a commercial company: Sony Corporation.

2.1. Please provide an amended Funding Statement declaring this commercial affiliation, as well as a statement regarding the Role of Funders in your study. If the funding organization did not play a role in the study design, data collection and analysis, decision to publish, or preparation of the manuscript and only provided financial support in the form of authors' salaries and/or research materials, please review your statements relating to the author contributions, and ensure you have specifically and accurately indicated the role(s) that these authors had in your study. You can update author roles in the Author Contributions section of the online submission form.

2.2. Please also provide an updated Competing Interests Statement declaring this commercial affiliation along with any other relevant declarations relating to employment, consultancy, patents, products in development, or marketed products, etc.  

Reviewers' comments:

Reviewer's Responses to Questions

**Comments to the Author**

1. Is the manuscript technically sound, and do the data support the conclusions?

Reviewer #1: Yes

Reviewer #2: Partly

Reviewer #3: Yes

2. Has the statistical analysis been performed appropriately and rigorously? 

Reviewer #1: Yes

Reviewer #2: N/A

Reviewer #3: Yes

3. Have the authors made all data underlying the findings in their manuscript fully available?

Reviewer #1: Yes

Reviewer #2: Yes

Reviewer #3: Yes

4. Is the manuscript presented in an intelligible fashion and written in standard English?

Reviewer #1: Yes

Reviewer #2: Yes

Reviewer #3: Yes

5. Review Comments to the Author

Reviewer #1: Major questions

General observations

The study presents relevant results to the Sports Science and for the area of human locomotion from a biological/evolutionary point of view, such as the fact that stride frequency is similar at the inflection point between groups (an innate pattern). In this sense, I suggest some observations:

- to modify the title to include some aspect of the result from a broader view and not just Sports Science

- in the introduction, this aspect needs to be considered in the justification / objectives / hypothesis (see line 13-14 of abstract: Since the influence of training on the basic V-C-S relationship is minimal, the basic pattern is largely innate). For example: What is this innate pattern??

- in the discussion, it is possible to intensify these questions by adding other arguments. For exemple: in Line 383, it was not clear, what are these neural mechanisms? What actually happens (brief) ? Another issue is that these neural mechanisms may be a physiological effect of an underlying mechanical mechanism (Line 461): the relationship between forward work (Wf) and vertical work (Wv) of center of mass and its effects on internal work - Wint (linked to lower limb movement) vs external work – Wext (linked to stride length). Slower running speeds Wext > Wint // faster running speeds Wext < Wint . Intersection speed (beetwen Wext and Wint) can match with critical point velocity (???) (see Cavagna et al. 2008 – Fig 3. (DOI: 10.1098/rspb.2007.1288))// and stride frequency effects on mechanics: Minetti and Saibene 1992 Mechanical work rate minimization and freely chosen stride frequency (SF) of human walking: a mathematical model // Minetti and Alexander 1997: A Theory of Metabolic Costs for Bipedal Gaits. // That is, innate factors need to be better discussed, or else to assume as limitations of the discussion

Methods

- It is necessary to make it clear how many times the same speed was performed, in addition, describe exactly how many speeds in relation to the maximum (%) were performed (Were there 10 differents speeds? And, How many times each?)

- Two different criteria were used to determine the time over the path? line 128 and 131. Please, to clarify and determine which of the two ways was used to determine the time.

- line 185: Variables except for running velocity and step length at maximal running velocity, and running velocity and step length at the inflection point exhibited non-normal distributions. Wouldn't it be better to describe which variables did not show normality?

Results

- A correlation of the spatiotemporal parameters was performed at Vmax. Why it was not done at the critical point speed? I think it would be really good point.

Discuss

- Important issues to highlight in the discussion:

Why SF is the same for all groups at the critical point velocity? Why around 60-65% of SF máx? Why stride length is different? At Vmax all groups were at maximum SL and SF, at critical point velocity not. The training history influenced SF and SL (futher) at maximum speed.

Line 314-316: it seems that the reasons for removing subjects 18 and 19 in the discussion were not the same as the results and methods line 222-224: “In addition, the data for two subjects showed no clear inflection point (sedentary subjects No. 18 and No. 19).”

Line 355: “Relative step length, however, still had a positive correlation with Vmax (Fig. 5F); the higher the Vmax, the higher the relative step length at the inflection point. In particular, in the range below the inflection point, subjects in the higher Vmax range (sprinters) showed a relative step length of close to 100%, indicating that they increased step length almost to the limit.” I don’t agree with that interpretation. In the first place, this correlation was weak (such as the speed at the inflection point) and it represents that regardless of the training history, all groups had a similar relative SL and quite close to the maximum SL.

Line 360: “This indicates that the higher the Vmax, the greater the scope for cadence increase in the velocity range above the inflection point.” It means that the history of the training influenced SF relative (not absolute) at the inflection point, that is, sprinters had at the inflection point a SF relative to the vmax lower than the others, although in absolute terms SF was the same, this interpretation was lacking.

Line 412 “Higher maximal velocity could be obtained with more efficient movements (running economy), and, thus, sprinters inevitably and unintentionally would pursue efficient movements.” I think the sentence should be written backwards due to the logic of the text and theoretical logic: runners running at a higher speed improve the efficiency and not the other way. Higher velocity needs more power and not economy.

- line 474: “Quantitative differences in the various parameters of the V-C-S relationship across the groups likely resulted from differences in innate ability and/or different running backgrounds.” I think it is necessary to highlight in the conclusion which parameters would be attributed to innate factors and which factors to running background, for example, speed running and SF at critical point (innate) and vmax (background).

Minor questions

- 1Graduate school of Sport Sciences: capital letter – school

- Line 446: extra space

Reviewer #2: Reviewers’ comments:

The purpose of the present study was to reveal how a change in running velocity altered the spatiotemporal adjustment between step length and cadence in subjects with different histories of engagement in running training. The main finding was that quantitative differences in the various parameters of the velocity-cadence-step length relationship across the groups likely resulted from differences in innate ability and/or different running backgrounds. The topic of this study is interesting! However, methodological and structural aspects compromise this submission. Below, some comments to improve the manuscript for that or future submission.

Keywords: It is suggested to adopt keywords that are not written in the title.

Abstract: The summary is apparently speculative. I suggest inform the numerical results and base the conclusion on the independent variables adopted.

Page 02, lines 04-06: Why the sedentary group ran 20m?

Page 02, line 09: The running velocity changed, but how did these changes occur in each group?

Page 02, lines 11-12: In fact, changes in running speed occur due to changes in length and then stride frequency (Nummela, et al., 2007).

Page 02, lines 12-14: Impossible to understand this sentence. Influence of training? Is this a variable independent of this study? I suggest removing this sentence.

Page 02, lines 16-17: I suggest inform the average values and standard deviations of running speeds, stride lengths and stride frequencies of each group.

Page 02, line 17: This cannot be said because no training has been investigated.

Page 02, lines 18-20: This conclusion is apparently speculative. Based in this abstract, it is not possible to conclude that, for example, the training history does not affect than cadence.

Introduction: The introduction is pertinent. However, the study's justification is not convincing. A more specific approach is suggested regarding the physiomechanical mechanisms of running (see, Minetti, A. E., for example). There is a need to adopt new bibliographic references. Finally, parts of the structure correspond to the methodology.

Page 03, lines 24-26: Please, cite a bibliographic reference.

Page 04, lines 49-51: It is not well defined that this adaptation may be due only to specific training of running. I suggest exploring this premise, based on physiomechanical studies. In addition, cite some references.

Page 04, lines 53-54: This was not introduced. What do studies show about it? Specifically, between stride length and frequency?

Page 04, lines 54-61: This is a methodological aspect (it should not be here).

Page 04, line 64: What is the concept of efficiency adopted here? Metabolic or Mechanical?

Page 04, lines 61-65: Based on which studies do the authors believe this?

Page 04, lines 66-71: This is a methodological aspect (it should not be here).

Methods: Some methodological questions are presented as a requirement for scientific acceptance.

Page 05, line 75: We know that there are physiomechanical differences between men and women runners. For example, the running economy. Was there any concern with sample homogenization in order to allow comparison between groups?

Page 05, line 77: The subjects were classified into 3 groups according to sports experience. However, it is known that physiomechanical changes in running occur for physiological reasons, too. The physical conditioning was homogenized between the groups? Also, what are the specific parameters (in particular background)?

Page 05, line 80: All

Page 05, line 82-84: Here I have some considerations to make. Some studies have demonstrated the influence of the sports motor gesture on the running technique. For example, studies with Triathlon. In addition, the way of carrying body mass can result in different physiomechanical behaviors of running (see allometric studies - Brisswalter, Kruel, etc ...). Why judo, volleyball, basketball, rowing, etc... All of these have different motor characteristics!

Page 06, line 95: Table 1: I believe that for the sampling characterization the maximal, aerobic, anaerobic capacities, and the leg length, should be informed. For age, I suggest adopt values without decimal places (mean and SD). Also specify the training volume.

Page 06, line 102: Why 30m?

Page 06, lines 102-103: Why 20m for the subjects of the sedentary group?

Page 06, line 109: Why 60 Hz? Normally, frequencies from 120 Hz are adopted, mainly in the case of sprinter runners...

Page 06, lines 110-112: Was any subjective adaptation of effort made prior to the performance?

Page 08, line 157: Figure 1: This is a result aspect (it should not be here). Also, there is no need for this information graphically.

Page 09, line 167: Figure?

Page 09, lines 165-173: This paragraph is not necessary.

Page 09, lines 176-178: Was a sample calculation performed? And a homogeneity test?

Page 09, lines 178-180: Why haven't parametric transformation tests been adopted? Non-parametric analyzes adopt median values, which can contribute to the existence of statistical errors.

Page 10, line 189: Please, adopt p in italics.

Results: There is a lot of information in each figure. Furthermore, the figures are not self-explanatory. Finally, the figures are not of good quality. I suggest that the authors adopt tables of results or detailing (in paragraphs) the main results of each set of results.

Page 10, lines 193-197: The Table 1 has already been presented. This information is redundant.

Page 10, lines 198-201: The Figure 1 has already been presented. This information is redundant. The quality of the figure is not good.

Page 10, line 202: Figure 2: I suggest adopting a data table or writing the results in a paragraph.

Page 11, line 219: These figures refer to a lot of information. It is suggested to adopt a paragraph highlighting the most important information.

Discussion: Very speculative.

Page 14, lines 292-294: Figure 2: Difficult to perceive this visually.

Page 14, lines 295-297: This changes in step length and cadence are justified why? A discussion is needed!

Page 15, lines 299-300: In this topic, there is no discussion of the results with the literature. There is a methodological justification! A physiomechanical methodological discussion of the results is not demonstrated.

Page 16, lines 332-333: The discussion is very descriptive and superficial. Physiomechanical aspects are not discussed, which could enrich the scientific contributions of the present study. An approach to neuromuscular and physiomechanical studies of running is suggested.

Pages 16-17, lines 344-348: This discussion is very superficial.

Page 19, line 392: This topic is very speculative.

Page 21, line 453: These limitations make the study quite superficial. Studies with a greater control of intervening parameters are suggested.

Page 22, line 467: There is no objective conclusion regarding the influence of sports motor characteristics (independent variables) on the dependent variables investigated in the present study. Again, there is a description of results.

Page 23, line 480: 11% references between 2021 and 2016; 59% between 2015 and 2000; 30% before 2015.

Reviewer #3: This manuscript presents interesting and novel findings. There may be broader implications, that can be investigated in the future, i.e. WHY is the critical velocity so consistent? What is being optimized? The study was simple but well-executed.

Major points:

1. Title: For such an interesting paper, the title is a bit boring, vague and not informative.

Suggestion: “No effect of running-specific training on the velocity at the step length-cadence inflection point”

2. Line 108 Can the authors provide any evidence that the subjects ran the 20/30 m at a constant velocity and were not accelerating/decelerating?

3. Line 106 “additional 10-30 meters” is too vague. Was it always 10, always 30, variable between subjects? Few if any sprinters reach maximum velocity in < 30m. this does not affect main conclusion but authors should note that max in this study is probably not maximum.

Minor Points:

Throughout the entire manuscript try to use “faster” or “slower” when describing velocity, not “higher” and “lower”

Perhaps PLOS One does not require it but I would be interested to know the contributions from each of the EIGHT authors. I am also curious as to why Sony is interested in this topic!

Line 6 Group average maximal running velocities ranked from fastest to slowest were: sprinters, distance runners….

Line 9 ‘as running velocity increased…”

Line 10 slower and faster than the inflection point,

Line 12 sentence beginning “Since” should be moved to the final sentence of Abstract. Also, add: “…is minimal, we surmise (or speculate or propose) the basic pattern is largely innate.”

Line 16 …the inflection point was not different between groups

Line 17 “frequency of running” here could be confused with cadence. I suggest: “Thus, the type, degree and frequency of running training had no effect.”

Line 24 suggestion, replace much of lines 24-30 with: “As a matter of fact, velocity equals the product of cadence and step length. But the relative contribution of each component to changing velocity differs across the velocity range. At slower velocities, speed is modulated primarily by adjusting step length whereas at faster velocities, speed is modulated more by changes in cadence (6).”

Line 34 “the relative contributions of cadence and step length optimize power production….”

Line 41+ is a mega paragraph. To communicate more clearly, I urge the authors to cut this into several smaller paragraphs, each with a topic sentence. E.g. break at line 51 “Therefore, the purpose…

Lines 44-47 can you provide a reference to support this statement?

Line 52 “investigate” rather than “reveal”

Line 56-61 This belongs in the Methods section

Line 66 …established the critical velocity…” not “point”

Line 80 … in many of the sports (comma) all subjects…

Line 83 capitalize American

Line 102 (only 20m for the sedentary…

Line 103 on which

Line 109 specify “video sampling frequency”

Line 119 …each trial ranged from 30 seconds to 5 minutes

Line 124+ again, this is a mega paragraph. You should break it into several smaller paragraphs

Line 129 end point (not goal point)

Line 130 cut (steps)

Line 131 cut (sec)

Line 133 end not “goal”

Line 134 also expressed as the dimensionless ratio (depicted = a picture)

Line 138 analyses (plural)

Line 167-168 I don’t understand. How can their data be inordinately low compared to their own data????

Line 287 be concise: The present study investigated the relative…

Or even better: We investigated the relative…

Lines 312-316 this is repeating what you already said earlier.

Lines 412=414 is too speculative. At least add “It may be” or ‘it might be possible that”

Line 418 I do not understand this claim.

Lines 421-438 I urge the authors to cut this rambling speculation.

6. PLOS authors have the option to publish the peer review history of their article (what does this mean?). If published, this will include your full peer review and any attached files.

Reviewer #1: **Yes: **Marcelo Coertjens

Reviewer #2: No

Reviewer #3: No

---

## [Author Response · Author response to Decision Letter 0]

29 Jun 2021

Dear reviewer and editor

I submitted it in the separated files (Cover letter and Response to reviewers).

I appreciate to cooperation.

---

## [Decision Letter · Decision Letter 1]

9 Aug 2021

PONE-D-21-04779R1

Spatiotemporal inflection points in human running: effects of training level and athletic modality.

PLOS ONE

Dear Dr. Kanosue,

Thank you for submitting your manuscript to PLOS ONE. After careful consideration, we feel that it has merit but does not fully meet PLOS ONE’s publication criteria as it currently stands. Therefore, we invite you to submit a revised version of the manuscript that addresses the points raised during the review process.

I agree with the reviewers that you did a good job of responding to the questions and suggestions they raised. However, some adjustments are still needed. I agree with all the points raised by reviewers 1 and 3, especially about the speculative content in the lines indicated by reviewer 3, and the need to report that individuals ran more than once at each intensity indicated by reviewer 1. Go ahead with this, you are already close to the publication. Certainly the paper will bring interesting insights into the basic kinematic relationships of human running.

We look forward to receiving your revised manuscript.

Kind regards,

Leonardo A. Peyré-Tartaruga, Ph.D.

Academic Editor

PLOS ONE

Journal Requirements:

Additional Editor Comments (if provided):

Reviewers' comments:

Reviewer's Responses to Questions

**Comments to the Author**

1. If the authors have adequately addressed your comments raised in a previous round of review and you feel that this manuscript is now acceptable for publication, you may indicate that here to bypass the “Comments to the Author” section, enter your conflict of interest statement in the “Confidential to Editor” section, and submit your "Accept" recommendation.

Reviewer #1: All comments have been addressed

Reviewer #2: All comments have been addressed

Reviewer #3: (No Response)

2. Is the manuscript technically sound, and do the data support the conclusions?

Reviewer #1: Yes

Reviewer #2: Yes

Reviewer #3: Yes

3. Has the statistical analysis been performed appropriately and rigorously? 

Reviewer #1: Yes

Reviewer #2: Yes

Reviewer #3: Yes

4. Have the authors made all data underlying the findings in their manuscript fully available?

Reviewer #1: Yes

Reviewer #2: Yes

Reviewer #3: Yes

5. Is the manuscript presented in an intelligible fashion and written in standard English?

Reviewer #1: Yes

Reviewer #2: Yes

Reviewer #3: Yes

6. Review Comments to the Author

Reviewer #1: Line 95-99: Subjects ran 30 times at speeds between 10-100%, so they ran about three times for each intensity (around that)!! That was my question in the previous comment. The suggestion only: I find it interesting to add that they ran more than once at roughly the same intensity – about 3x (even if it wasn't exactly the same as it was subjective. It is no a problem).

Line 321: Please, to review this phrase: “Scatter plots of all subjects of all the groups showed only a weak correlation between the Vmax and the velocity at the inflection point (Fig. 4D).” In fact, the weak correlation was the velocity at the inflection point normalized with Vmax, since absolute velocity had strong correlation (Fig 4A)

Line 32: space (“maximum .”)

Line 138: space (“Deming regression .)

Reviewer #2: The changes made have substantially improved the manuscript. The authors agreed that the previous version contained too many speculative descriptions and also that the structure was too complicated. So, I recommend the publication of this manuscript in PLOS ONE.

The purpose of the present study was to investigate how a change in running velocity changes the spatiotemporal adjustment between step length and cadence in subjects with different histories of engagement in running training. The main finding is that the inflection point appeared at a similar cadence and at similar a relative velocity, across all groups. These results imply that the influence of running-specific training on the inflection point is minimal.

The authors agreed that the previous version contained too many speculative descriptions and also that the structure was too complicated. The Fig. 2 (which showed the V-C-S curve of all subjects in a diagram) was excluded as the basic messages overlapped with Fig. 3 and the supplementary figures. Was added Table 2 to provide numerical data on the inflection point. Finally, the order of Fig. 3 (new Fig. 3) and Fig. 4 (new Fig. 2) has been changed. The changes mentioned, accompanied by the changes in the text, made the study acceptable for publication.

Abstract: The authors have modified the abstract so as to be minimally speculative. It is suggested that references are not cited (page 2, lines 15 and 17).

Introduction: This topic has been substantially improved and the authors also clarified some issues. In fact, authors have not done any physiological or biomechanical measurement in this study, just focusing on velocity, SF, and SL. The introduction presents relationships between justification, problem and scientific objective.

Methods: The changes made, resulting from some suggestions from the reviewers, improved the understanding of the methodological aspects adopted. The main focus of the present study was the “running-specific training to run faster", which is the specific background considered.

Results: The results were properly adjusted in the text. Authors have revised the explanation of each figure to make then all as clear as possible. Was added Table 2 to provide numerical data concerning the inflection point.

Discussion: The modifications were pertinent. Now, I am confident about the quantitative analysis of the present study. I am, too, sure that this study will provide a good foundation for future studies that analyze the neuronal/physiomechanical mechanisms that are involved.

Limitations of the present study: Very important!

Conclusions: Pertinent!

References: Authors have added some references.

Reviewer #3: The manuscript is much improved.

My only major suggestion is to delete Lines 350-379. I find this all too speculative. The study itself is solid and obviously the investigators are very excited to explore the topic further. But, in my opinion, the excessive speculation without data detracts rather than positively adds to the manuscript.

Minor

Line 11 consider “critical velocity” rather than critical point

Line 14 This pattern was commonly observed in all four groups. not only in sprinters and distance runners, as has already been reported (Weyand et al., 2000; Nummela et al., 2007), but also in active athletes and sedentary individuals.

Line 16 This pattern may reflect an energy saving strategy (Yanni & Hay, 2004).

Line 19 … wide variety of athletic experience of the subjects…

Line 47…. In running training. Namely, we studied: 1. sprinters, 2. distance runners…

Line 53 slower velocity ranges

Line 54. Therefore, we hypothesized: 1. the running step length/cadence patterns of individuals would be influenced by their running training experience and overall physical activity levels and 2. Distance runners would…

Line 63 They were assigned into one of four groups depending on their current/pervious running training.

Line 67 Although running is involved in many sports, all subjects…

Line 72 without a history

Line 86 all-weather

Line 100 match the exact percentage

Line 102 When running at the minimum velocity, subjects followed…

Line 103 slowly

Line 104 interval between trials ranged

Line 113 The instant at which…

Line 114 identified from the position

Line 115 number of steps by the time taken to cover that distance. The number of steps…

Line 118 between the instant of first foot

Line 123 30 trials by each subject

Line 124 was designated as their maximal running velocity. (cut: of the subject)

Line 125 start new paragraph here

Line 162 Next, post-hoc pairwise…

Line 189 …as the intersection point

Line 192 significant differences between the groups in terms of maximum running velocity

Line 193 .. revealed that the maximal velocity of the sprinters was faster compared to all the other subject groups…

Line 196 The distance runner group exhibited significantly faster…

Line 198 …absolute step length and step length normalized to height.

Line 207 open circles indicate each individual subject

Line 212 both absolute velocity and velocity normalized to maximal running velocity

Line 255 Table 2 Kinematic variables at the inflection point

Line 266 approximately constant

Line 274 … (Vmax) and: running velocity (A), cadence (B)…

Line 288 As expected, compared to the sprinters, maximal running velocities were progressively slower in the distance runner, active athlete and sedentary groups.

Line 293 Among the subject groups, the sprinters were the tallest and the sedentary group was the shortest.

Line 294 … with Vmax was well-preserved …

Line 297 and longer steps

Line 299 It appears that a slower cadence would have required …

Line 302 In all four subject groups, an abrupt… Table 2). Velocity changes below the inflection point occurred mainly by modulating step length and velocity changes above the inflection point occurred mainly via cadence modulation.

Line 307-309 cut this if you adopt my suggestions for line 302+

Line 310 …inflection point has a significant

Line 314 start a new paragraph here: Overall, regardless of training history…

Line 316 inflection not inflexion. Yes, they sound the same, but English is a weird language.

Line 316 remained

Line 330 …relationship are common across different subject groups

Line 331 …could be related to quality/quantity…

Line 335 …be expected for the…

Line 338 cut: “and performing”

Line 339 It seems reasonable that some portion…

Line 340 start new paragraph

Line 340 point also follows

Line 343 I don’t think that study actually MEASURED energy, they just postulate it

Line 381 Sub-heading could be “Future Studies”

Line 382-386 delete

Line 388 I don’t understand this part about “modeling”

7. PLOS authors have the option to publish the peer review history of their article (what does this mean?). If published, this will include your full peer review and any attached files.

Reviewer #1: **Yes: **Marcelo Coertjens

Reviewer #2: No

Reviewer #3: No

---

## [Author Response · Author response to Decision Letter 1]

24 Aug 2021

Dear editor and reviewers

I submitted it in the separated files (Cover letter and Response to reviewers).

I appreciate to cooperation.

---

## [Decision Letter · Decision Letter 2]

30 Sep 2021

PONE-D-21-04779R2Spatiotemporal inflection points in human running: effects of training level and athletic modality.PLOS ONE

Dear Dr. Kanosue,

Thank you for submitting your manuscript to PLOS ONE. After careful consideration, we feel that it has merit but does not fully meet PLOS ONE’s publication criteria as it currently stands. Therefore, we invite you to submit a revised version of the manuscript that addresses the points raised during the review process.

Congrats, you have done a good work until now. As you can see below, just one reviewer has some minor points. However, I have some further suggestions to improve the clarity of text and figures. Also, I feel that your discussion is somewhat empty of rationale and references. Please, consider following carefully my recommendations as follows. Please, pay special attention to your two abstracts (in the system and into the manuscript), the version in the system is not updated. In the first paragraph of introduction, consider including this work: https://pubmed.ncbi.nlm.nih.gov/34197674/ line 33 - consider including this reference in the list (5,11,12): https://doi.org/10.3389/fphys.2018.01789 line 39 - here, you uses references that didn't investigate that (specially 16). Consider including this classical work about this topic: https://pubmed.ncbi.nlm.nih.gov/9305998/ . Even, if you look carefully the figures of Cavagna et al., 1997, you'll see that 3Hz or very close (in slower speeds) at self-selected cadences. line 342 - Consider including this reference showing that the stride length was associated with better running economy in distance runners (https://pubmed.ncbi.nlm.nih.gov/22978185/) line 363 - consider including this review where the concept of efficiency is broadly explained (https://doi.org/10.3389/fphys.2018.01789). line 364 - I suggest this writing:This and also fatigue (https://pubmed.ncbi.nlm.nih.gov/26214838/), aging (https://pubmed.ncbi.nlm.nih.gov/18077249/ and https://pubmed.ncbi.nlm.nih.gov/27116643/) and gender (https://pubmed.ncbi.nlm.nih.gov/33064810/) differences, if any, are topics that merit future analysis. 

Figure 1 - consider removing the decimal scale (for example 4 instead 4.0) in the horizontal and iso-speed scales.

Figure 2 - consider removing the decimal scale (for example 4 instead 4.0) in the vertical (A) and hor (B,C,D) scales.

Figure 3 - consider removing the decimal scale (for example 4 instead 4.0) in the horizontal scale.

Figure 4 - consider removing the decimal scale (for example 4 instead 4.0) in the vertical and hor scales.

We look forward to receiving your revised manuscript.

Kind regards,

Leonardo A. Peyré-Tartaruga, Ph.D.

Academic Editor

PLOS ONE

Journal Requirements:

Reviewers' comments:

Reviewer's Responses to Questions

**Comments to the Author**

1. If the authors have adequately addressed your comments raised in a previous round of review and you feel that this manuscript is now acceptable for publication, you may indicate that here to bypass the “Comments to the Author” section, enter your conflict of interest statement in the “Confidential to Editor” section, and submit your "Accept" recommendation.

Reviewer #1: All comments have been addressed

Reviewer #2: All comments have been addressed

Reviewer #3: (No Response)

2. Is the manuscript technically sound, and do the data support the conclusions?

Reviewer #1: Yes

Reviewer #2: Yes

Reviewer #3: Yes

3. Has the statistical analysis been performed appropriately and rigorously? 

Reviewer #1: Yes

Reviewer #2: Yes

Reviewer #3: Yes

4. Have the authors made all data underlying the findings in their manuscript fully available?

Reviewer #1: Yes

Reviewer #2: Yes

Reviewer #3: No

5. Is the manuscript presented in an intelligible fashion and written in standard English?

Reviewer #1: Yes

Reviewer #2: Yes

Reviewer #3: Yes

6. Review Comments to the Author

Reviewer #1: (No Response)

Reviewer #2: The changes made have substantially improved the manuscript.

I recommend the publication of this manuscript in PLOS ONE.

Reviewer #3: I have mostly minor comments, no need for me to review again (3rd time!?).

Major

Lines 208 and 209 I think you mean Figures 2, not 4!

Liner 218 I think you mean Figure 3A not 2A

Minor

Line 31 velocity approaches the maximum

Line 56 in the slower

Line 61 add two commas: backgrounds, in terms of their running experience, participated

Line 69 add colon. were: soccer

Line 71 in their sport

Line 93 differed between trials

Line 114 The instants….. end points were

Line 157 SPSS Statistics

Line 159 test. Maximal running…

Line 160 Plural. …to have non-normal distributions.

Line 161 data for maximal

Line 165 investigate the possible mechanisms responsible for the inflection point

Line 175 was set at p < 0.05

Line 221 the velocities between

Line 301 step lengths

Line 319 inflection not “inflexion”

Line 346 runners increased velocity by elongating both absolute step length (Figure 4C) and relative step length (Figure 4F) all the way to the upper running speed limit.

Line 349 Obviously, maximal velocity…

Line 350 A faster velocity

Line 351 Sprinters need to develop

Line 359 could help to answer our overall question.

Line 363 Is the V-C-S pattern innate or does it develop…

Line 364 also sex differences..

Line 370 in the slower and faster

7. PLOS authors have the option to publish the peer review history of their article (what does this mean?). If published, this will include your full peer review and any attached files.

Reviewer #1: **Yes: **Marcelo Coertjens

Reviewer #2: No

Reviewer #3: No

---

## [Author Response · Author response to Decision Letter 2]

3 Oct 2021

We submitted it as "Response to Reviewers"

---

## [Editor Report · Decision Letter 3]

5 Oct 2021

Spatiotemporal inflection points in human running: effects of training level and athletic modality.

PONE-D-21-04779R3

Dear Dr. Kanosue,

We’re pleased to inform you that your manuscript has been judged scientifically suitable for publication and will be formally accepted for publication once it meets all outstanding technical requirements.

I am sure that editorial process has helped to turn out the paper clearer and better. Congrats, very good paper!

Kind regards,

Leonardo A. Peyré-Tartaruga, Ph.D.

Academic Editor

PLOS ONE

---

## [Editor Report · Acceptance letter]

8 Oct 2021

PONE-D-21-04779R3 

Spatiotemporal inflection points in human running: effects of training level and athletic modality. 

Dear Dr. Kanosue:

I'm pleased to inform you that your manuscript has been deemed suitable for publication in PLOS ONE. Congratulations! Your manuscript is now with our production department. 

Kind regards, 

on behalf of

Professor Leonardo A. Peyré-Tartaruga 

Academic Editor

PLOS ONE